# Multilevel Regulation of Membrane Proteins in Response to Metal and Metalloid Stress: A Lesson from Yeast

**DOI:** 10.3390/ijms25084450

**Published:** 2024-04-18

**Authors:** Kacper Zbieralski, Jacek Staszewski, Julia Konczak, Natalia Lazarewicz, Malgorzata Nowicka-Kazmierczak, Donata Wawrzycka, Ewa Maciaszczyk-Dziubinska

**Affiliations:** Department of Genetics and Cell Physiology, Faculty of Biological Sciences, University of Wroclaw, 50-328 Wroclaw, Poland; kacper.zbieralski2@uwr.edu.pl (K.Z.); jacek.staszewski2@uwr.edu.pl (J.S.); 321911@uwr.edu.pl (J.K.); natalia.lazarewicz@uwr.edu.pl (N.L.); malgorzata.nowicka@uwr.edu.pl (M.N.-K.); donata.wawrzycka@uwr.edu.pl (D.W.)

**Keywords:** membrane transporters, transporter regulation, arsenic, antimony, cadmium, *Saccharomyces cerevisiae*, transporter-related diseases

## Abstract

In the face of flourishing industrialization and global trade, heavy metal and metalloid contamination of the environment is a growing concern throughout the world. The widespread presence of highly toxic compounds of arsenic, antimony, and cadmium in nature poses a particular threat to human health. Prolonged exposure to these toxins has been associated with severe human diseases, including cancer, diabetes, and neurodegenerative disorders. These toxins are known to induce analogous cellular stresses, such as DNA damage, disturbance of redox homeostasis, and proteotoxicity. To overcome these threats and improve or devise treatment methods, it is crucial to understand the mechanisms of cellular detoxification in metal and metalloid stress. Membrane proteins are key cellular components involved in the uptake, vacuolar/lysosomal sequestration, and efflux of these compounds; thus, deciphering the multilevel regulation of these proteins is of the utmost importance. In this review, we summarize data on the mechanisms of arsenic, antimony, and cadmium detoxification in the context of membrane proteome. We used yeast *Saccharomyces cerevisiae* as a eukaryotic model to elucidate the complex mechanisms of the production, regulation, and degradation of selected membrane transporters under metal(loid)-induced stress conditions. Additionally, we present data on orthologues membrane proteins involved in metal(loid)-associated diseases in humans.

## 1. Introduction

Heavy metals and metalloids are ubiquitous in environmental compartments such as the Earth’s crust, water, air, and sediments. In nature, they can be found at volcanic sites, in geothermally active areas, and in natural deposits [1,2,3,4]. However, the contamination with these elements originates also from anthropogenic sources, such as the metallurgic industry, mining, fossil fuel extraction, and global transport [5,6,7] (Figure 1). Importantly, they have also found use in various industrial applications such as the production of batteries, alloys, and ceramics but also as potential drugs in therapies against human diseases [7,8,9]. Nevertheless, prolonged exposure to toxic elements such as arsenic, antimony, and cadmium has been numerously shown to cause a multitude of serious human diseases and disorders (Figure 1).

The compounds of highly toxic elements such as cadmium, arsenic, and antimony share some common mechanisms of cellular toxicity. For instance, they can interact with various cellular macromolecules, such as DNA, proteins, and membranes. As a result, they induce serious cellular disturbances, including DNA damage [10,11,12], oxidative stress [13,14,15], and proteotoxicity [16,17]. To cope with these challenges, in the course of evolution, cells have developed distinct mechanisms of detoxification and adaptation involving the regulation of specific transporters and enzymes involved in the uptake, sequestration, metabolism, or export of these toxic compounds. Cells need to adjust the membrane protein composition, as they function as a contact site between the extracellular and intracellular environment (plasma membrane, PM), a synthesis hub for newly produced membrane proteins (endoplasmic reticulum, ER), or a sequestration site of toxic compounds (vacuolar membrane, VM) (Figure 2). The production, activity, and stability of membrane proteins are tightly regulated at multiple levels, including transcription (through the induction of specific activators and repressors as well as translational machinery), but also at the protein level by various post-translational modifications (e.g., phosphorylation and ubiquitylation) (Figure 2). These modifications can affect the localization, conformation, interaction, and degradation rate of membrane proteins, thus modulating their function and efficiency in the process of uptake, metabolism, and efflux of toxic compounds.

Considering these facts, the knowledge of precise control of transporters and the proteins regulating them at all stages—from transcription control through post-translational modifications to degradation—is of paramount importance regarding the devising and improvement of future therapies against metal(loid)-associated diseases and metal(loid)-based drugs. In this work, we summarize the knowledge regarding the molecular mechanisms responsible for the regulation of membrane proteins involved in the detoxification of arsenic, antimony, and cadmium, as well as the importance of these proteins for cell functioning and, consequently, the impact of disorders of these processes and proteins on human health. We focus on the data obtained in studies carried out using yeast *Saccharomyces cerevisiae*, which is a powerful model organism for inquiring about the molecular basis of the response to cellular stress. Yeast shares many basic cellular processes, mechanisms, and proteins with humans [18]. Additionally, the mechanisms of metal and metalloid detoxification in yeast are relatively well-described in relation to other eukaryotic organisms, including humans. They are also somewhat resistant to toxic agents, surviving a comparably deadly dosage of arsenic or antimony, counted in milli-molar concentrations [10,19]. Those dissimilarities in sensitivity may result from various differences, including the presence of the cell wall, simpler metabolic requirements, as well as other, additional systems like specific transporters, which are not present in the mammalian cells. Therefore, this organism is a useful tool for investigating the processes occurring in cellular stress response, as well as their further implications for human health. Thus, the intention of this review is to highlight and recapitulate the available literature on the topic of several selected yet crucial arsenic, antimony, and cadmium yeast transporters and the protein factors influencing their expression, post-translation regulation, and degradation. Additionally, it links the disorders of the transporter orthologues with human diseases and more. 

### 1.1. The Toxicity of Arsenic Compounds

Arsenic (As) naturally occurs in four oxidation states: −3, 0, +3, and +5. The compounds in the +3 and +5 oxidation states constitute a vast majority of arsenic forms in the environment and living organisms. The forms in the +3 oxidation state [(As(III)] prevail in oxygen-free conditions with reducing characteristics (such as rocks or river mud), while the forms in the +5 oxidation state [As(V)] dominate oxygen-rich environments, such as surface waters or soil [20]. In general, inorganic trivalent arsenic species are more toxic than inorganic pentavalent ones. At the same time, organoarsenic compounds are considered even more potent toxins, and aromatic arsenicals are considered the most toxic arsenic species [21]. Inorganic arsenic compounds can be biotransformed into organic forms in cells by forming complexes with glutathione or by methylation [3,22]. Moreover, they can bind biologically active molecules and form their arsenic derivatives (e.g., arsenocholine, arsenobetaine, arsenolipids), which can be integrated into distinct macromolecules or compete with crucial substrates in metabolic pathways, disrupting or completely blocking their functioning [22,23]. For instance, As(V) structurally resembles PO_4_^−^ ions; thus, it can inhibit enzymes that involve phosphate, such as glyceraldehyde-3-phosphate dehydrogenase [24].

Arsenic is known to induce a variety of cellular stresses. The high affinity of trivalent arsenic species to sulfhydryl groups allows their binding to cysteine residues in proteins, which may result in protein misfolding or the inhibition of enzymatic activity [16,25]. Taking into account that thiol binding is stronger within adjacent cysteine residue sites, in particular, this results in the displacement of zinc ion cofactors from the zinc finger domains, mainly of C3H1 and C4 zinc finger type [26,27,28]. Therefore, many crucial DNA binding proteins, including members of the DNA repair systems, may undergo a direct inhibition of function upon arsenic exposure, especially in conditions of zinc deficiency [29,30]. Moreover, arsenicals have been found to negatively affect the cytoskeleton [31] and strongly induce oxidative stress [31,32]. Although, initially, the genotoxic activity of arsenic has been thought to depend on ROS-induced DNA damage, Litwin et al. demonstrated that arsenic also induces oxidative stress and transcription-independent DNA breakage [33]. Inorganic as well as organic methylated arsenic forms were also found to impact growth signaling in epidermal cells, which may further increase the risk of skin cancer development [34,35].

### 1.2. The Toxicity of Antimony Compounds

Similarly to arsenic, antimony (Sb) is a metalloid that belongs to group 15 of the periodic table and, therefore, presents the same range of oxidation states from −3 to +5. Analogously to arsenic, the most common form of antimony in oxic environments is +5, while in anoxic environments it is +3 [36]. It can also form organoantimonials (e.g., methylated species), which are considered the least toxic, while antimonites (Sb(III)) are the most toxic ones [3].

Although the toxic properties of antimony are not fully understood, there is evidence that antimonials affect cells similarly to arsenic species (Figure 1). It has already been demonstrated that antimony can bind proteins [17] and induce oxidative stress [14], as well as direct and indirect DNA damage [11]. Like arsenic, it is also similarly potent in altering signaling pathways in epidermal cells [37,38]. These observations indeed confirm the cytotoxic properties of antimony species; nevertheless, further studies are required to fully uncover the mechanisms of antimony toxicity.

### 1.3. The Toxicity of Cadmium Compounds

Cadmium (Cd) is a heavy metal that naturally occurs in the Earth’s crust and atmosphere; however, similarly to metalloids, arsenic, and antimony, no biological functions of cadmium in higher organisms have been found [39] with widely known toxic traits [40]. Cadmium contamination comes from the burning of fossil fuels and mining [41]. This metal serves as a model toxicant for triggering cellular stress responses specific to heavy metals. It is known to induce heat-shock and oxidative stress responses, e.g., inducing heat-shock-related proteins like HSP70 or heme oxygenase (HMOX1) [42,43]. While most of cadmium’s biological effects are related to its ability to alter the cellular redox state, some effects may be due to structural similarities between cadmium and calcium or zinc [1].

Cadmium-induced cellular oxidative stress is likely caused by the disruption of redox homeostasis related to the mishandling of redox-active metals. This leads to lipid and protein oxidation and oxidative DNA damage. Cd has only one oxidation state (Cd^2+^); therefore, it cannot directly generate free radicals. However, it has been reported that cadmium can cause indirect generation of various radicals, including the superoxide radical, hydroxyl radical, and nitric oxide [44,45]. In addition to its direct effects on cellular redox balance, cadmium competes with calcium for ion channels in cell membranes and with calcium-regulated proteins inside cells due to molecular mimicry of calcium ions. Cadmium not only competes with calcium channels for entry but also stimulates them, leading to excessive accumulation of calcium in different cellular compartments [46]. As a result, it may induce apoptotic cell death [46,47].

Moreover, studies indicate that many structural components of ribosomes, translation initiation factors, and elongation factors are downregulated by Cd(II) treatment. However, some proteins involved in the ubiquitin-dependent protein metabolic process and 19S proteasome regulatory subunits were upregulated. These results suggest that the regulation of translation and protein degradation is also a crucial part of the cellular response to cadmium stress [48]. Further, cadmium can alter the structure of membrane proteins. It has the most significant effect on newly formed, partially folded membrane proteins that are prone to misfolding and aggregation [1,49].

## 2. Transcriptional Regulation of Membrane Protein Genes in Response to Metal and Metalloid Stress

Cell survival requires rapid adaptation to constant environmental changes. Although stress conditions impose rapid changes in the existing protein composition, cells also have to tightly regulate their transcription profiles (for the summary of the transporters and their regulators described below, see Table 1). In the presence of toxic compounds, it is crucial to simultaneously block the transcription of genes coding undesirable proteins (e.g., transporters involved in the uptake of the stressor) and upregulate those essential for survival (e.g., enzymes involved in the metabolism of the toxin) (Figure 2).

The key cellular components controlling the selective binding of RNA polymerases to DNA are transcription factors (TFs) [50]. Determined groups of TFs cooperate in response to given conditions, acting as activators or repressors for various groups of genes. In yeast, the response to heavy metal and metalloid stress conditions involves the activity of two major groups of TFs—the representatives of the yeast activator protein (YAP) family and the activator proteins involved in pleiotropic drug resistance (PDR) [51,52]. 

The proteins of the YAP family are transcription activators resembling the activation protein (AP)-1 factors in humans [53]. They are stress response-related transcription factors of the JUN, FOS, MAF, and ATF protein families [54]. The YAP proteins are basic leucine-zipper (bZIP) domain-containing transcription factors, which bind specific DNA sequence motifs in the form of homo- or heterodimers. Moreover, all representatives of the family involved in metal/metalloid detoxification, including Yap1p, Yap2p, and Yap8p, harbor two unique Cysteine-Rich Domains (n-CRD and c-CRD), which contain several evolutionarily conserved cysteine residues, essential for the function of these proteins [55,56]. Together, these TFs condition the expression of a multitude of membrane proteins involved in arsenic, antimony, and cadmium detoxification.

The PDR gene network, on the other hand, comprises various proteins involved in general drug response [51]. Among functionally and structurally distinct proteins in the network, the PDR TFs (e.g., Pdr1-3p) regulate the transcription of a wide array of regulatory and membrane proteins. These TFs recognize and bind to the DNA motifs called PDR elements (PDREs), which are commonly present upstream of transcription start sites in their target genes. These include genes coding various ATP-binding cassette (ABC) transporters (e.g., Pdr5p, Pdr10p, Pdr15p, Snq2p, and Yor1p), as well as major facilitator (MFS) transporters (e.g., Rsb1p, Rta1p, Hxt2p, Tpo1p, Hxt9p, Hxt11p, Flr1p, Atr1p, and Sge1p) [51,57].
ijms-25-04450-t001_Table 1Table 1Membrane proteins involved in arsenic, antimony, and cadmium response and their human homologs and regulators.Yeast Membrane Protein(s)Human Structural and/or Functional HomologsSubcellular LocalizationToxic Metal/Metalloid Inducing ResponseNotable Transcription RegulatorsNotable Post-Translational RegulatorsAcr3p (↑)N/DPMAs(III), As(V), Sb(III)Yap8p [58]Rsp5p, Art3p, Art4p [59]Fps1p (↑/↓)AQP9PMAs(III), As(V), Sb(III)Fhl1p [60], Gcn5p, Med2p, Stp1p [60]Hog1p, Rgc1, Rgc2 [61,62]Hxt1-7p (↓)GLUT1-5PMAs(III)Rgt1p, Mig1p [63]Rsp5p, Art4p (HXT1/3/6) [64,65],  Art7p (HXT3/6) [64,65], Art8p (HXT2/3/6/7) [64,66], Atg1 (Hxt1p) [67],  Cdk1 (Hxt1p) [67,68],  Npr1p (Hxt1p, Hxt3p) [69]Pho84p (↓)SLC34/SLC20PMAs(V), As(III)?Pho4p [70], Spt7p [71]Pho86p [72], Cdc28p [68], Rsp5p [73]Ftr1p (↓)N/DPMAs(V), As(III)Aft1-2p [74]Vta1p [75]Fet3p (↓)N/DPMAs(V), As(III)Aft1p [74]Vta1p [75]Vba3 (?)N/DVMAs(V)Fkh1p, Put3p, Tfc7p, Yap6p [76]Tul1p [77]Pca1p (↑)ATP7AER, PMCd(II)Spt10p [78], Msn2p [76], Gcn4p [79]Ubc6/7p, Doa10p, Cue1p [80]Yor1p (↑)MRP/CFTR-type ABC transporters/ABCC12PMCd(II)Pdr1/3p [81]Rsp5p, Art1-5/7p [82]Ycf1p (↑?)MRP/CFTR-type ABC transportersVMAs(III), Sb(III), Cd(II)Yap1p [83]Tul1p [77], Cka1p [77,84]Zrt1p (?)SLC39A1-3PMCd(II)Zap1p [85],  Aft1/2p? [74]Rsp5p [86]Bpt1 (?)ABCC6VMCd(II)Gln3p, Pho2p,  Rtg3p [60]Atg1p [69]Vmr1 (?)ABCC10VMCd(II)Msn2p, Msn4p,  Gcn4p [87]N/DPM—plasma membrane, VM—vacuolar membrane, ER—endoplasmic reticulum, ↑—upregulation, ↓—downregulation, ?—ambiguous/unknown regulation, N/D—no data.

### 2.1. Transcription of Genes Coding Membrane Proteins Related to Arsenic and Antimony Stress

Arsenic and antimony compounds are potent toxins that heavily affect cellular transcription profiles. For instance, arsenic stress has been demonstrated to negatively regulate the Sfp1 transcription activator involved in the transcription of ribosomal genes [88]. On the other hand, it stimulates general stress-responsive transcription activators Msn2 and Msn4 [88]. Other transcription regulators that were significantly upregulated in arsenic stress include Rpn4p, Fhl1p, Yap1p, Yap2p, Pre1p, Hsf1p, and Met31p [89]. There is little data concerning the differences in the regulation of transcription between arsenic and antimony in yeast. The available transcriptomic data mostly come from the studies on transcription profiles of the *Leishmania* parasites responsible for leishmaniasis in humans, which is treated mostly with antimony-based drugs [90]. Indeed, it has been demonstrated that antimony-resistant *Leishmania* strains display significant changes in transcriptomic profile in comparison to the control strains, including pronounced changes in the expression of multiple membrane protein-coding genes [91].

In yeast, the key membrane protein involved in the detoxification of arsenicals is the PM transporter Acr3 (Figure 2). It is a member of the bile/arsenite/riboflavin transporter (BART) superfamily and a founding member of the arsenic compound resistance (Acr)-3 family, which is ubiquitously present in prokaryotes, fungi, and plants [92,93,94]. Acr3p acts as an antiporter, which utilizes the proton gradient generated by the cell membrane H+-ATPase to extrude As(III) out of the cell (Figure 2) [95]. It also cooperates with the *ACR2* gene, which encodes an arsenate reductase that catalyzes the reduction of As(V) to As(III) [96]. Acr3p is predominantly regulated at the transcription level. Under normal conditions, the transcription of the *ACR3* gene is turned off; however, exposure to arsenic has been demonstrated to strikingly induce *ACR3* transcription [95]. Importantly, both the *ACR3* and *ACR2* genes are localized closely on the yeast chromosome XVI and share the same bi-directing promoter sequence [58,97]. The transcription of both *ACR3* and *ACR2* depends on a single TF Yap8p [58]. It has been demonstrated that Yap8 activation depends on the Hog1 kinase, which is a homolog of mammalian mitogen-activated protein kinase (MAPK) p38 [98]. Consistently, the deletion of the *HOG1* gene results in reduced transcription of the *ACR3* gene [99]. Recently, a zinc finger domain-containing protein Etp1 has been found to affect the transcription of *ACR3* [100]. Although Etp1 has been found to interact with Yap8p, the Etp1-dependent regulation of *ACR3* has been demonstrated to occur independently of Yap8p [100]. Strikingly, Yap8p has also been observed to be post-translationally regulated by arsenic itself. In the absence of arsenic, the protein is continuously degraded by ubiquitin-dependent proteolysis [101]. However, Yap8p directly binds As(III) independently of any other yeast protein and is stabilized in effect, thus acting as an arsenic sensor [102].

The aquaglyceroporin Fps1 of the major intrinsic protein (MIP) family is a membrane protein crucial for arsenic and antimony transport across the PM (Figure 2) [103]. Chromatin immunoprecipitation (ChIP) data indicate that under normal conditions, the transcription rate of the *FPS1* gene may be influenced by transcription regulators Fhl1p, Gcn5p, Med2p, and Stp1p [60,76]. Arsenic exposure, however, causes a rapid inhibition of *FPS1* transcription [19]. Moreover, it has been proposed that the 5′UTR uOFR region present in the transcript of *FPS1* may limit its translation rate, preventing deleterious effects of unregulated *FPS1* expression under normal conditions [104].

As trivalent arsenic can adapt a form of ring-like structure mimicking glucose, the hexose transporter (HXT) family of PM permeases has been found to serve as another uptake pathway for arsenicals in yeast (Figure 2) [105]. Similarly, homologous transporters of the glucose transporter (GLUT) family in humans have been demonstrated to transport As(III) as well [106]. Under normal conditions, the transcription of yeast *HXT* genes is regulated by glucose availability. It has been proposed that in the absence of glucose, the *HXT1-3* genes coding low-affinity hexose transporters are repressed by the Rgt1 transcription factor, while the *HXT6-7* genes coding high-affinity glucose transporters remain unrepressed. Following the increase in glucose availability, Rgt1p disassociates from the *HXT* gene promoters, also allowing the transcription of low-affinity glucose transporters. High levels of glucose, on the other hand, inhibit the transcription of *HXT2,6-7* genes through the Mig1-dependent repression mechanism. Under these conditions, the transcription of *HXT3* remains active, while the transcription of *HXT1* is induced by Rgt1p, which acts as a transcription activator in this context [63]. Arsenic treatment seems to not affect the transcription of at least several *HXT* genes, as the mRNA levels of *HXT2,6-7* insignificantly change in the presence of arsenic [107]. As under these conditions, the protein level of these transporters rapidly decreases, the HXT transporters seem to be regulated at the protein level instead [107].

Due to its structure resembling inorganic phosphate, pentavalent arsenic enters the cells through several phosphate (PHO) transporters (Figure 2). The main PHO transporter involved in this process is the high-affinity transporter Pho84p [70,108,109]. Normally, the transcription of *PHO84* is regulated by the basic helix-loop-helix (bHLH) TF Pho4, which belongs to the myc-family [110]. Moreover, it has been proposed that the transcription of *PHO84* is also coupled with nutrient-sensing signaling pathways [111]. Additionally, the expression of *PHO84* was suggested to be repressed by the antisense transcription of the gene [112]. Recently, however, it has been proposed that the 3‘UTR region of the transcript, rather than the antisense RNA, is responsible for the downregulation of the transporter [112,113].

Arsenate treatment has also been demonstrated to affect the high-affinity iron permease Ftr1 and ferroxidase Fet3 (Figure 2) at both transcriptional and protein levels. Interestingly, it does so in an opposing manner. Although As(V) induces the degradation of Ftr1p/Fet3p, it also strongly stimulates the Aft1-2 (Activator of Ferrous Transport 1 and 2) TFs, activators of the Fe regulon. The high expression of *FTR1* and *FET3* has been proposed to be a cellular response to iron deficiency and does not translate to high protein levels. The obtained data indicate that the *FTR1* and *FET3* mRNA is rapidly degraded by Xrn1p—a nuclease involved in mRNA decay. As the strain devoid of the *FTR1* and *FET3* genes displays a phenotype of high tolerance to arsenate, this high-affinity uptake system has been speculated to participate in arsenate uptake [114], and the observed phenomena may be a cellular strategy to reduce arsenate toxicity.

Given the structural and biochemical similarity between trivalent arsenic and antimony, most of the membrane proteins associated with antimony response overlap with the arsenic-related ones. For instance, Sb(III) treatment has been shown to stimulate the transcription of *ACR3* in yeast, although this phenomenon is much less pronounced than in the case of arsenic treatment [115]. The main TF involved in antimony response in yeast is Yap1p. Complex microarray studies of strains overexpressing Yap1p indicated 17 genes with at least a three-fold increased expression [53]. Yap1p regulates the transcription rate of three important yeast ABC transporters, Ycf1p, Snq2p, Pdr5p [83], as well as two transporters of the MFS family, Atr1p and Flr1p [116], and various genes involved in the biosynthesis of thioredoxin and glutathione [117]. The most important antimony resistance-related transporter gene regulated by Yap1p is the *YCF1*, which encodes a vacuolar C-type ABC (ABCC) transporter homologous to the human MRP and CFTR transporters [118,119]. Ycf1p actively transports the glutathione-antimony conjugates (Sb(GS)_3_) into the vacuole, strongly contributing to yeast tolerance to antimony (Figure 2) [19,58,118,120,121,122]. The regulation of *YCF1* expression is not well understood. For instance, *YCF1* is transcribed under normal conditions as well as stress conditions. The *ycf1*Δ strain displays a hypersensitivity phenotype to antimony, and the overexpression of the Yap1p has been found to promote the transcription of the *YCF1* gene. However, the metalloid treatment seems to not affect the expression of *YCF1*; thus, the complex mechanism of the regulation of *YCF1* transcription remains elusive [58,123,124].

### 2.2. Transcription of Genes Coding Membrane Proteins Related to Cadmium Stress

Similarly to arsenic and antimony, exposure to cadmium significantly affects the transcriptomic profile of yeast. A high-throughput RNAseq analysis revealed that the plasma membrane protein-encoding cadmium-responsive genes were strongly affected by cadmium treatment [125]. Another RNAseq analysis revealed that the Gpp2, Tec1, and Sfg1 TFs, as well as PM transporters Hxt5, Yct1, and Ptr2, might be regulated by the Hog1p signaling pathway in response to cadmium treatment [125,126]. Moreover, it was proposed that TFs Hot1, Msn2, and Msn4 might negatively regulate the expression of PM cysteine permease *YCT1* [126]. One of the ABC transporters involved in the detoxification of cadmium, Vmr1p (Figure 2), is also regulated at the transcriptional level by the Msn2p and Msn4p, as well as starvation-responsive Gcn4p [87].

The YAP family members are also important regulators of the cell response to cadmium stress. Yap1p regulates the transcription of the gene coding the Ycf1 transporter, which is responsible for vacuolar sequestration not only of Sb(GS)_3_ but also Cd(GS)_2_ and As(GS)_3_ [127,128]. The Yap2 TF, otherwise known as Cad1p (cadmium resistance 1), shares the highest homology with Yap1p. However, regardless of similarities in their function and overlapping gene targets, Yap1p and Yap2p are not redundant TFs [129]. The overexpression of Yap2p has been found to confer resistance to cadmium [129]. This corresponds well with the fact that Yap2p can directly interact with cadmium, which stimulates the transactivating potential of the factor [130]. It has also been demonstrated that Yap2p is regulated by the MAPKAP (MAPK-activated protein) kinase Rck1, which negatively regulates the protein half-life and nuclear accumulation [131]. Together, Yap1p and Yap2p cooperate in the cellular response to cadmium, providing the expression of genes necessary for survival in cadmium and cadmium-related oxidative stress [130].

The PM ABCC transporter Yor1 (Figure 2), which is essential for cellular response to cadmium in yeast, is related to the MRP/CFTR group of ABC transporters, such as the human MRP1 and yeast Ycf1 proteins [132]. It is a PM glutathione conjugates transporter facilitating the export of Cd(GS)_2_ out of the cell [132,133]. The expression of Yor1 is regulated mostly by the PDR factors Pdr1-3 [134,135,136,137].

A P-type ATPase Pca1 is another crucial cadmium exporter (Figure 2) [138,139]. The available data indicate that the transcription of *PCA1* may be influenced by several factors, including Spt10p [78], Msn2p [76], and Gcn4p [79]. However, post-translational stabilization rather than increased transcription levels was found to be a key mechanism regulating the activity of the protein [80].

The PM high-affinity zinc importer Zrt1, on the other hand, has been found to facilitate cadmium uptake in yeast (Figure 2). The transcription of *ZRT1* is controlled by the Zap1 transcription factor in response to changing intracellular zinc levels [140]. Zap1p binds to the zinc-responsive elements (ZREs) in the *ZRT1* promoter in response to low zinc availability [85]. Although the mechanism regulating the expression of *ZRT1* in cadmium stress is ambiguous, it is known to be repressed by the negative regulators Mot3p and Rox1p in response to osmotic stress [141].

## 3. Post-Translational Regulation of Membrane Proteins in Response to Metal and Metalloid Stress

### 3.1. General Effects of Metal and Metalloid Exposure on the Regulation of Membrane Proteins

Apart from the transcript regulatory alterations previously discussed, proper post-translational modifications are frequently essential for the proper functioning of a particular transporter, from its synthesis and sorting to its ultimate removal from the membrane. A growing amount of research demonstrated that functional changes in cells, which go beyond variations in protein abundance, are linked to modifications in the state and structure of significant protein regulatory factors as well as transport proteins themselves. These changes arise from disruptions in their regulatory or post-translational processing.

The most common examples of regulatory processes are phosphorylation-induced changes and their consequences for the cellular proteome, which occur in all organisms, from yeasts to humans [142,143]. Due to the complexity of these processes, many of them, as well as their molecular basis in yeast, have not yet been fully described and require additional study. For example, by comparing the yeast transcriptome, proteome, and phosphoproteome, as well as looking at phosphorylation states, it was possible to decipher how genetic effects alter signaling networks. Thus, it has been presented that phosphorylation properties are closely related to cell physiological parameters, such as chemical resistance or cell morphology, compared to transcript or protein abundance [144]. Another piece of evidence was the description of several specific phosphorylation states and sites. They were correlated with several stress resistance traits in the context of a novel, high-quality quantitative trait loci (QTL) multiomics technique in yeast. For the first time, this demonstrated the central importance of protein phosphorylation in the adaptation of stress responses in living organisms (Figure 2) [144]. It is also important to acknowledge that the level of phosphorylation also translates into disorders of post-translational processes in cancer and their impact on molecular pathways and cellular processes [145]. Therefore, protein regulation and its post-translational modifications are an important element of survival under stress conditions in various species [146,147]. However, little is known about detailed studies of protein state changes in response to toxic agents.

Exposure to toxic metals and metalloids affects transcriptional and post-translational modifications of not only the stress-related membrane proteins but also of their regulators themselves. An example of this would be the upregulation of autophagy pathways, which, in addition to the proteasome, play an important role in controlling the number of proteins in the cell. Proteomic studies demonstrated a broad upregulation of autophagy components at the protein level. One of the proteins whose level increased the most in response to arsenic stress was Atg8p, a crucial autophagy regulator required for the production of autophagosomes. On the other hand, as a result of the presence of arsenic, the downregulation of components of the ribosomal machinery has also been reported. The results show that a significant decrease in the amount of more than half of the subunits was observed after arsenic treatment [148]. Similarly, many important proteins undergo quantitative changes in the presence of factors such as cadmium. A significant number of ribosome structural components (Rpl702p, Rpl3001p, Rps2p), translation initiation factors (Tif33p, Tif211p), and elongation factors (Tef5p) are downregulated upon Cd(II) treatment. Importantly, in the case of the proteome regulation machinery, the increase in abundance levels of some of the proteins involved in ubiquitylation processes (Ubx4p) or proteasomal degradation (the 19S proteasome regulatory subunits Rpn502p, Rpn11p, Rpt6p, and Mts4p) was observed. This indicates the importance of degradation pathways in the presence of cadmium in the cell. It has also been stated that the multilayer regulation of pathways critical for Cd(II) tolerance in *S. pombe* is regulated by Spc1p and Zip1p. Zip1p is believed to be essential for the primary regulation of important sulfur metabolism-related enzymes and is required for cadmium detoxification, whereas Spc1p is crucial for acute reactions to cadmium stress [48].

### 3.2. Regulation and Post-Translational Modifications of Arsenic and Antimony Transporters

One of the best-described regulatory transporters involved in arsenic transport in yeast is Fps1p. This transporter contains a mitogen-activated protein kinase (MAPK) phosphorylation site (Thr231) in its long cytosolic-facing N-terminal tail, which is crucial for gating [149,150,151]. The deletion of this residue or the entire N-terminal domain results in increased sensitivity to As(III) and Sb(III) due to high levels of unregulated metalloid influx [19,151]. It has been demonstrated that the mitogen-activated protein kinase Hog1 mediates this phosphorylation. Hog1p directly and negatively controls Fps1p-mediated transport by phosphorylating Thr231. As(III) and Sb(III) activate Hog1 kinase, cells missing Hog1p (*hog1*Δ) are particularly sensitive to both metalloids and exhibit higher rates of Fps1p-dependent As(III) absorption [103,151]. Additionally, in the regulation of the Hog1p phosphorylation level, two other positive regulators of Fps1p activity have been identified, Rgc1p and Rgc2p/Ask10p, which are pleckstrin homology (PH) domain proteins. As(III) tolerance is increased by the inactivation of Fps1p by the deletion of *RGC1* or *RGC2* [152]. Fps1p forms a homotetramer, and a redundant pair of regulators Rgc1p and Rgc2p govern the activity of this channel. Rgc1p and Rgc2p bind to the C-terminal cytoplasmic domain of Fps1 to keep it in the open channel state. Hog1p phosphorylates Rgc1p and Rgc2p, which removes these regulators from Fps1p and ultimately closes the transporter channel. Rgc1p and Rgc2p have been demonstrated to form both homodimers and heterodimers with each other. The N-terminal domain of Rgc2p mediates the formation of dimers, and mutations that inhibit Rgc2p dimerization impede its capacity to open Fps1 [61,62]. Furthermore, it has been demonstrated that methylated arsenite—MAs(III)—is a strong inhibitor of the protein tyrosine phosphatases (Ptp2p and Ptp3p), which normally maintain the inactive state of Hog1p. Inhibition of Ptp2p and Ptp3p by MAs(III) leads to increased Hog1p phosphorylation without the activation of protein kinases that act upstream of stress-activated MAPKs (SAPKs). Furthermore, unlike As(III), arsenate [As(V)], a pentavalent form of arsenic, also activates Hog1p, but it does so by activating Hog1p through MAP/ERK kinase (MEK) Pbs2 [153].

Unlike Fps1p, yeast Acr3p, in terms of arsenic and antimony transport, is regulated mainly at the transcriptional level. But yet, a comprehensive investigation of the domains and regions in charge of Acr3p’s appropriate metalloid transport was carried out in addition to identifying the structures in charge of Acr3p’s removal from the membrane. It was determined that the mobile transport domain consists of two transmembrane (TM) regions, the TM3-5 and TM8-10, while the scaffolding domain, on which the transport domain glides, is made up of the TM1-2 and TM6-7 domains. The conserved areas of TM4, TM5, TM9, and TM10 were characterized, including TM9’s G353 residue, which might assist with substrate binding and is required for Acr3 transport activity, as well as TM4’s C151 residue, which may act as a metalloid binding site during translocation and is necessary for the As(III) and Sb(III) antiport by Acr3 in yeast. Furthermore, as shown, the V173A and E353D Acr3 mutants are unable to export the Sb(III) and As(III) out of the cell, respectively [154]. 

Taking into account the regulation of Hxt1-7 transporters involved in arsenic transport, their regulation is also controlled at multiple different pathways. Hxt1p has been shown to be phosphorylated in vitro via the Atg1 kinase [67], while in vivo via the Cdk1 kinase [67,68], and Npr1 kinase included in TORC1-dependent feedback control [69]. In the case of the Hxt2p, it was shown that the peptide fraction in the mass spectrometry study is enriched in phosphorylation more than 25-fold in the case of Rad53 kinase deletion [155,156]. The same approach as presented for the discovered Hxt1p phosphorylation site also proposed Npr1 kinase-dependent phosphorylation of Hxt3p [69]. Regarding the Hxt5 transporter, in addition to several studies describing novel phosphorylation sites using the global proteomic studies [68,157,158], another post-translational regulatory modification through the succinylation of the lysine residue has been presented [159]. In the case of Hxt7p, phosphorylated residues were identified thanks to high-throughput proteomic techniques. Importantly, the same studies also contributed to the discovery of phosphorylation sites of the other hexose transporter family members [68,158,160]. Interestingly, from this group of proteins, the least characterized is the Hxt6 transporter, whose regulators have not yet been described.

Similarly, for other arsenic response proteins, such as Tat1, Frt1, Fet3, and Pho84, multiple phosphorylation sites have been proven in the previously mentioned genome-wide phosphoproteomic studies [68,155,157,158,160,161]. What is also worth noting, some of the studies may suggest the involvement of the TOR-controlled pathway in the regulation of proteins such as Tat1, and at the same time, the precise mechanism and exact regulators still require thorough investigation [162]. Additionally, in the case of Tat1p, one residue of succinylation-mediated lysine modification was also confirmed [163]. For the Pho84 transporter, according to data obtained by Holt and colleagues, the phosphorylation of this protein can be driven in the Cdk1-dependent pathway [68]. The Fet3p also has multiple N-glycosylation sites described thanks to the quantitative profiling and mapping studies [164,165].

### 3.3. Regulation and Post-Translational Modifications of Cadmium Transporters

The post-translational regulation of Ycf1p occurs at the levels of intracellular trafficking, phosphorylation, and proteolytic processing by Pep4 protease [166,167,168,169]. The phosphorylation of the ABC core domain, as well as the guanine exchange factor Tus1, both positively regulate Ycf1p [170]. The phosphorylation of residues S908 and T911 in its core ABC domain, driven by the Tus1p, positively regulates Ycf1p activity. At the same time, compared to wild-type Ycf1p, the S251A mutant shows higher cadmium resistance in vivo and increased Ycf1p-dependent [(3)H]estradiol-beta-17-glucuronide transport in vitro. Thus, it is proposed that S251 phosphorylation negatively regulates Ycf1p activity. Moreover, Ycf1p function increases upon the deletion of two kinase genes, *CKA1* and *HAL5*, which were discovered by the integrated Membrane Yeast Two-Hybrid (iMYTH) screen. Taking into account these results, as well as additional genetic tests, it was confirmed that the Cka1 kinase may directly or indirectly phosphorylate S251 to control Ycf1p activity [169]. Nevertheless, only a little Cd sensitivity results from altering the phosphorylated residues or eliminating *TUS1* [168].

The biogenesis of the Yor1 transporter depends on the level of transport from the endoplasmic reticulum (ER) via the secretory pathway, similar to other proteins in the ABC transporter family. Two DXE element-like sequence motifs commonly found in other ER exit proteins are necessary for Yor1p to be transported from the ER to its site of function in the plasma membrane. The protein’s function is lost when the N-terminal DXE fragment is eliminated. Therefore, these findings highlight the significance of the signals linked to this domain in the proper control and sorting of the protein to the membrane; the removal of this domain potentially results in the mislocalization and, ultimately, degradation of mutational protein [171]. Using Stable Isotope Labeling by/with Amino acids in Cell culture (SILAC)-based experiments, a comprehensive phosphoproteome screening for budding yeast was presented. Among over 30,000 phosphosites detected under DNA-damaging conditions and/or before arrest in various cell cycle states, six new phosphorylation sites were identified for Yor1p [158]. Moreover, Swaney et al. demonstrated four additional sites, as well as new ubiquitylation sites [160]. Although a significant number of phosphorylation sites have been identified for Yor1p, knowledge about the mechanisms of phosphorylation of this transporter remains scarcely understood so far [157,161,172]. It has been suggested that Yor1p contains one putative phosphorylation site targeted by the Hog1 kinase. However, at the same time, it should be taken into account that in the absence of detected evidence of the physical interaction of Hog1p and Yor1p, further studies are required [173]. Noteworthily, a study investigating phosphorylation targets of the Cdk1 kinase identified a phosphorylation site also for Yor1p as a possible substrate [68].

The Zrt1 transporter’s activity is controlled through the mechanisms regulating both its transcription [85,140] and vacuolar degradation [174]. Specific glycosylation sites have been identified for this transporter using mass spectrometry-based mapping techniques [165,175]. As in the case of the Yor1 transporter, thanks to high-throughput experiments, it was also possible to confirm various phosphorylated residues of the Zrt1 protein [68,158,160,172].

As in the case of many previously mentioned transporters involved in arsenic transport, in the case of the Vmr1p [158], Bpt1p, and Ypk9p [68,155,157,158,160,161], numerous proteomic data confirmed multiple phosphorylation sites of these proteins. At the same time, the involvement of the Atg1 and Slt2 kinases [69] as at least one of the kinases involved in the phosphorylation of the Bpt1p and Ypk9p, respectively, is indicated.

## 4. Degradation of Membrane Proteins in Response to Metal and Metalloid Stress

The ability to adjust the protein composition of cellular membranes is a fundamental asset, allowing the survival of cells in response to ever-changing environmental cues. The rapid and specific response to stress conditions requires not only the synthesis of new membrane proteins necessary for survival but also the turnover of damaged, dispensable, and undesirable ones (Figure 2). In eukaryotic cells, the degradation of soluble and membrane proteins is mostly regulated by ubiquitylation, which is a post-translational modification consisting of the covalent binding of a small protein ubiquitin (Ub) to the acceptor lysine residues in the substrate [176] in distinct combinations of polyUb chains [177]. For instance, the UbK48-type chains target proteins for proteasomal degradation, whereas the UbK63-type chains target membrane proteins for degradation in vacuoles/lysosomes [178]. 

The degradation of membrane proteins is distinctly regulated in different cellular compartments. At the ER, membrane proteins are downregulated by the endoplasmic reticulum-associated degradation (ERAD) machinery [179]. In yeast, it includes the ER membrane-embedded ubiquitin ligases Doa10 and Hrd1 (homologous to human ligases MARCHF6 and SYNV1, respectively), which tag their substrates with the K48-type polyUb chains [180,181], thus targeting them for proteolysis in the proteasome [179,182]. On the other hand, the degradation of proteins present at the PM occurs mainly through their ubiquitylation-dependent endocytosis, endosomal sorting, and subsequent vacuolar/lysosomal degradation [183]. In both yeast and animal cells, ubiquitylation acts as a signal-inducing endocytosis [184,185,186], and the main ligases responsible for the ubiquitylation of PM proteins belong to the Rsp5/NEDD4 family [187], which bind their substrates in cooperation with the adaptor proteins of the α-arrestin family [188]. These ligases tag their PM substrates with the K63-linked polyUb chains, which are recognized by the highly conserved endosomal sorting complexes required for transport (ESCRT) [189,190,191]. The ESCRT machinery is crucial for the sorting of membrane proteins to the endosomal lumen, and their proteolysis occurs in the vacuolar/lysosomal lumen after the fusion of endosomes and vacuoles/lysosomes [189]. Both the Rsp5/NEDD4 ligases and the ESCRT complexes, together with ubiquitin ligases Pib1p and the defective SREBP cleavage (Dsc) complex, are also involved in the degradation of membrane proteins at the vacuolar/lysosomal membrane [77,192]. 

### 4.1. Degradation of Membrane Proteins in Response to Arsenic and Antimony Stress

Recent proteomic studies demonstrate that exposure to arsenic induces intensive remodeling of the proteome in both yeast and human cells [148,193], including changes in the membrane proteome composition. As it was mentioned before, the proper response to arsenicals and antimonials in yeast requires their export out of the cell mainly through the arsenic/antimony Acr3 transporter [115,154]. Recently, the process of the degradation of the protein has been examined. It has been established that Acr3p is a moderately stable protein, and its half-life is not related to the presence of arsenic [59]. Acr3p has been found to undergo vacuolar proteolysis dependent on α-arrestins Art3 and Art4, which are speculated to bind the negatively charged N-terminus of the transporter and promote its Rsp5-dependent polyubiquitylation and degradation [59].

Both arsenic and antimony are substrates for the aquaglyceroporin Fps1 [19,103]. Although the adjustment of the activity of Fps1p is quite well understood, the regulation of its half-life under metalloid stress remains mostly uncharacterized. The activity of this channel depends on the phosphorylation by the stress-response-related MAPK kinase Hog1, which, under distinct conditions, regulates the activity and/or half-life of Fps1p. For instance, in the presence of high levels of acetic acid, Hog1p rapidly phosphorylates the T231 and S537 residues of Fps1p, effectively targeting it for endocytosis and degradation [194]. In arsenic stress, on the other hand, Fps1p seems to be regulated in terms of switching between the open/closed channel states rather than protein stability [103,148]. Nevertheless, a high-throughput proteomic study demonstrated a slight decrease in Fps1p level after prolonged exposure to 1 mM As(III) [148]. Given that Fps1p is a main uptake pathway for As(III) and Sb(III), this phenomenon may be connected to a long-term strategy of cellular adjustment to metalloid stress consistent with the observations that lack of *FPS1* or maintaining the channel in a closed state increases yeast tolerance to As(III) and Sb(III) [19,103].

Exposure to arsenic has been found to rapidly downregulate the yeast transporters of the hexose transporter (HXT) family (the glucose transporters GLUT in humans) [107], which is another entrance pathway for As(III) [105,106]. While the mid/high-affinity hexose transporters Hxt2/6/7 have been observed to degrade most rapidly, the protein levels of the low-affinity transporters Hxt1/3 decreased as well, and the level of the stress conditions-related Hxt5 transporter has not changed [107]. In yeast, arsenite acts as a competitive substrate for the HXT transporters [105]. The ability of arsenic to disturb proper glucose metabolism poses serious energetic stress for cells; hence, the downregulation of this arsenic import pathway seems to be an effective mechanism for cellular protection. The degradation of the HXT and GLUT transporters involves the Rsp5/NEDD4 ubiquitin ligases and several α-arrestin adaptor proteins. Although there is little information on the exact mechanism of arsenic-dependent degradation of these transporters, it has been demonstrated that in arsenic stress, several HXT transporters are degraded in an Rsp5p- and K63-type polyUb chain-dependent manner [107].

It is worth mentioning that exposure to As(III) has also been found to trigger the degradation of a multitude of nutrient transporters in yeast, such as the arginine permease Can1, lysine permease Lyp1, and multi-amino acid permease Tat1 [148]. However, the methionine permease Mup1 has been shown to be upregulated in response to As(III) instead [148]. Moreover, the overexpression of the yeast vacuolar amino acid permease Vba3 has been recently observed to provide the tolerance of yeast cells to arsenate, although the mechanism responsible for the phenomenon seems to not be related to increased As(V) accumulation in the vacuole [72]. In eukaryotes, the TORC1 kinase complex is the master regulator of nutrient response. When active, TORC1 orchestrates the degradation of the amino acid permeases [195,196]. Surprisingly, though, arsenic has been previously demonstrated to inhibit TORC1 [88]. Thus, these observations suggest a more complex mechanism of the arsenic-dependent regulation of nutrient transporters which is yet to be elucidated. Given the lack of sufficient data, the full unraveling of the relationship between arsenic stress and nutrient transporters requires further studies.

The resistance to both arsenic and antimony can be acquired by their sequestration into the vacuolar lumen by the ABC transporters such as Ycf1p. The regulation of the stability of this protein is scarcely characterized. A recent structural study revealed that Ycf1p requires its main phosphorylation sites (S908 and T911) in order to maintain structural stability [119]. These sites are located in the R-domain of Ycf1p, which is a functionally conserved region of interaction between the ABCC transporters and various protein kinases, and the disruption has been demonstrated to cause high instability and the rapid degradation of Ycf1p [119]. Several ubiquitylation sites in Ycf1 have been identified so far [160,197]. Interestingly, the K504 and K862 residues of Ycf1p have been described as modified with the K63-type polyUb chains [197]. The Dsc complex subunit ubiquitin ligase Tul1 has been demonstrated to promote vacuolar degradation of Ycf1p when overexpressed, suggesting it may be the ligases responsible for its ubiquitylation [77]. At the same time, it has been demonstrated that Tul1p provides tolerance to arsenate when overproduced [72]. Nevertheless, the relationship between the phosphorylation and degradation of Ycf1p remains elusive. It is worth mentioning, however, that upon arsenic exposure, the protein level of Ycf1p seems to remain virtually unchanged [148,197]. 

As for the influx of pentavalent arsenicals, the main proteins responsible for the process are inorganic phosphate transporters, especially the high-affinity phosphate transporter Pho84 [72,109]. Its PM localization is regulated by the Pho86 protein, which conditions proper ER-exit of Pho84p and increases the PM level of Pho84p when overexpressed [72]. Little is known, however, about the regulated degradation of Pho84p, especially in response to arsenic stress. High concentrations of inorganic phosphate have been demonstrated to trigger the degradation of Pho84p in a protein kinase A (PKA)-dependent manner [198]. Although there is no sufficient data on the degradation rate of Pho84p in the presence of arsenate, the *pho84*Δ strain is hypertolerant to As(V) [199], and a high-throughput proteomic analysis revealed that in response to arsenite, the protein level of Pho84p indeed rapidly decreases [148]. Moreover, Pho84p has been found to physically interact with Rsp5p, which may indicate its role in the degradation of Pho84p [200]. Nevertheless, the exact mechanism regulating the degradation of Pho84p in response to arsenicals remains ambiguous.

Pentavalent arsenic also downregulates the high-affinity iron uptake system involving the Ftr1 and Fet3 proteins. The *FTR1* and *FET3* mRNA, as well as the Fet3 protein, are rapidly degraded upon arsenate treatment, and this phenomenon is related to a lower accumulation of arsenic in yeast cells [114]. However, As(III) treatment also causes a decrease in the protein levels of Fet3p and Ftr1p [148]. Whether this effect is similarly caused by both As(III) and As(V) is not clear, and the cellular reduction of arsenate to trivalent arsenic cannot be excluded, especially after prolonged exposure time. Nevertheless, these data strongly imply that there exists an important overlap between arsenic and iron uptake and metabolism pathways.

### 4.2. Degradation of Membrane Proteins in Response to Cadmium Stress

A highly toxic heavy metal, cadmium, has been demonstrated to cause strong disbalance in redox and divalent-ion homeostasis [15,46,201,202,203]. Cd(II) is also able to affect signaling pathways in the cell. For instance, cadmium not only is a competitive substrate for calcium channels but also activates them, providing means for the hyperaccumulation of calcium in distinct cellular compartments and dysregulating calcium-dependent signaling pathways [46,47]. Cadmium is also able to hijack other divalent ion uptake pathways. For instance, the zinc PM transporter Zrt1 has been demonstrated to participate in Cd(II) import to the cytosol. In the presence of high concentrations of Zn and Cd, Zrt1p is removed from the cell surface to prevent the uptake of toxic Cd and excess Zn [174,204]. Zrt1p inactivation involves Rsp5p-dependent ubiquitylation, followed by endocytosis and degradation in the vacuole [86,204].

Cadmium is characterized by its severe proteotoxic activity. The acute cadmium-induced proteotoxic stress is connected to Cd(II)-induced protein misfolding, as Cd(II) has been demonstrated to bind thiol groups, effectively disrupting disulfide bonds between cysteine residues [205,206]. Cadmium has been found to affect the structure of not only cytosolic [207] but also membrane proteins, especially at the ER [208]. For instance, Cd(II) has been demonstrated to target ER proteins, and cadmium exposure strongly upregulates the ERAD pathway [209].

Similarly to arsenic and antimony, cadmium can be either sequestered into the vacuolar lumen or extruded out of the cell. The former process is facilitated mostly by the ABCC transporter Ycf1 and, to some extent, its paralogues Bpt1p and Vmr1p [124], as well as several transporters of other divalent metals (e.g., Ypk9p, Zrc1p) (Figure 2) [210,211]. As for the efflux of Cd(II), several proteins capable of exporting Cd(II) out of the cell were identified. For instance, Cd(II) is one of the substrates for the PM multidrug transporter Yor1p [133], which requires α-arrestins Art1-5/7 to be properly degraded in response to cycloheximide treatment [82,133]. However, the most interesting example of PM Cd(II) exporters is the P-type ATPase Pca1 of *S. cerevisiae* (Figure 2). Intriguingly, under normal conditions, the ATPase is constantly degraded in the ER through the ERAD machinery [80]. In the absence of cadmium, ubiquitin ligase Doa10p recognizes the hydrophobic degradation-promoting sequence within the N-terminal region of Pca1p [80,212]. In effect, Doa10p ubiquitylates Pca1p in cooperation with ubiquitin-conjugating (Ubc) enzymes Ubc6/7 and targets Pca1p for proteasomal degradation. Strikingly, upon cadmium exposure, the cysteine-rich N-terminal tail of Pca1p directly binds Cd(II), effectively preventing Doa10p from recognizing the degradation signal [212]. It results in ER rescue and subsequent transport to the PM, where Pca1p participates in Cd(II) export [80]. Altogether, this phenomenon provides a pool of unmatured Pca1p, which can be instantly re-localized from the ER to the PM in the presence of Cd(II), allowing a rapid response to cadmium stress.

## 5. The Role of Membrane Transporters in Metal- and Metalloid-Related Human Pathologies

Heavy metals and metalloids ubiquitously occur in the environment. Hence, all organisms developed mechanisms for their detoxification, which often involve similar membrane transporters and channels. Many yeast proteins involved in the transport of arsenic, antimony, and cadmium have similar counterparts to those found in humans. Importantly, arsenic, antimony, and cadmium not only have malignant effects on human health but also found use in medicine and various areas of industry (Figure 1). In this chapter, we present the impact of arsenic, antimony, and cadmium on human health and disease pathogenesis in the context of selected human membrane proteins (Table 2).

### 5.1. Health Implications of Impaired Functioning of Arsenic and Antimony Transporters in Humans 

Arsenic is an especially dangerous contaminant, as more than 200 million people live in areas with elevated levels of this element (mostly in Asia and Latin America) [229,230]. Arsenic is a known carcinogen and neurotoxin. It leads to skin, bladder, liver, and lung cancer [229]. It is harmful to the skin, causing numerous diseases and pigmentation changes [231]. Exposure to arsenic has also been linked to diabetes mellitus due to arsenic-induced pancreatic β-cell death [232]. Additionally, arsenic exposure seems to be associated with insulin resistance and cardiovascular disorders as a result of impaired vascular response to neurotransmitters and abnormalities in vascular muscle calcium signaling [233]. A proper diet rich in vitamins and natural antioxidants has been shown to counter arsenic toxicity [234]. There are also attempts at arsenic detoxification using nutraceuticals [235]. Despite its deleteriousness, arsenic-based therapeutics can be used to treat some serious diseases, such as acute promyelocytic leukemia (APL) and lung cancer [236].

Antimony is a metalloid commonly used in the metal and plastic industries. Sb enters the human body mainly through the inhalation of contaminated air. The uptake from the gastrointestinal tract is lower than 1% [237]. Sb exposure has been linked to pulmonary toxicity (pneumonitis), causing chronic inflammation, mild fibrosis, and elevated cancer risk [238]. Moreover, Sb accumulates in red blood cells due to integration with hemoglobin [239] and leads to hemolysis [240]. Sb alters serum cytokine and immunoglobulin levels [241], lowers thyroid hormone levels [242], and induces chromosome aberrations [238]. Sb has also been shown to affect human reproduction, as it decreases sperm concentration [243]. During pregnancy, Sb disrupts blood glucose homeostasis [244] and induces hypertension [245]. Sb also has a positive effect on human cells. It has been shown that Sb induced the apoptosis of acute promyelocytic leukemia cells [246]. Antimony is also widely used as an antileishmanial drug [9]. 

Several human membrane transporters have been shown to transport arsenic and its compounds. ATP-binding cassette (ABC) proteins, such as ABCA1, ABCB1, and ABCC1, are linked to acquired arsenic and drug resistance [247]. ABCB1 and ABCC1 upregulation leads to decreased arsenic accumulation and, thus, higher resistance [248]. Global deficiency of ABCA1 causes Tangier disease, in which one of the symptoms is an almost complete loss of high-density lipoprotein cholesterol (HDL), as well as splenomegaly, enlarged tonsils, and atherosclerosis [213]. ABCA1 is also involved in coronary heart disease (CHD), type 2 diabetes (T2D), thrombosis, age-related macular degeneration (AMD), glaucoma, viral infections, and neurological disorders, such as traumatic brain injury, Alzheimer’s disease, and Parkinson’s disease [214]. ABCB1 is also overexpressed in ovarian cancer [215], causing multidrug resistance, which is a significant difficulty encountered during chemotherapy [247]. 

Glucose transporter 1 (GLUT1) has been found to facilitate arsenic uptake in humans [106]. Permease GLUT1 is mainly expressed in erythrocytes and epithelial cells of the blood-brain barrier. Its physiological role is to mediate the transport of glucose into the brain cells across the blood-brain barrier [249]. Thus, GLUT1 catalyzes the majority of arsenic uptake into erythrocytes and the brain, possibly leading to arsenic-induced cardiovascular disorders and neurotoxicity [106]. The downregulation of GLUT1 results in GLUT1-deficiency syndrome (GLUT1DS), the symptoms of which include delayed neurological development and various neurological disorders [216]. 

It was shown that Human Aquaporin 9 (AQP9, homologous to yeast Fps1p) also contributes to arsenic uptake [250]. AQP9 is expressed mainly in the liver and in leukocytes [251]. Alterations in AQP9 expression result in various diseases, leading especially to liver injury and immune disorders, but also inflammation and numerous cancers, contributing as a promising therapeutic target and biomarker [252]. Improper AQP9 expression promotes chronic liver injury (CLI), which is a common disease resulting in hepatic steatosis, liver fibrosis, and eventually, if not treated, hepatocellular carcinoma (HCC) [217]. In CLI, AQP9 is overexpressed [217], while in HCC, AQP9 is downregulated [219]. Using arsenic-sensitive and arsenic-resistant liver cancer cell lines, it has been established that the phosphorylation of AQP9 regulated by p38 kinase (homologous to yeast Hog1p) regulates cellular arsenic sensitivity [253]. Such tolerance can limit arsenic-dependent cancer therapies; thus, the explanation of the mechanism underlying the resistance is an important step for improving therapeutic strategies [254]. As AQP9 is one of the most common AQPs in immune cells, it is closely related to the regulation of immune response [255]. Its expression increases after lipopolysaccharide (LPS) influence, playing an important role in the development of early stages of LPS-induced endotoxic shock and indicating AQP9 as a promising drug target in sepsis treatment [256]. The overexpression of AQP9 is also connected with systemic inflammatory response syndrome (SIRS) [218]. Moreover, it regulates the migration of immune cells, facilitating their motility and chemosensing [218,257]. AQP9 abnormalities are also connected to male and female infertility and pregnancy complications [258]. In females, AQP9 is downregulated in polycystic ovary syndrome (PCOS) and is associated with hyperandrogenism [220]. In males, a lower expression level of AQP9 alters sperm maturation and storage [221]. 

There are no confirmed antimony transporters in humans, but there is some research indicating that membrane proteins involved in arsenic uptake (AQP9, ABC transporters) also facilitate antimony transport [259], suggesting the involvement of similar transporters and pathologies related to them as in the case of arsenic.

### 5.2. Health Implications of Impaired Functioning of Cadmium Transporters in Humans

Cadmium is not transitioned into less toxic compounds and is ineffectively eliminated in humans. Moreover, due to its exceedingly long biological half-live reaching 10–30 years, it accumulates easily, especially in the kidneys and liver, effectively causing organ failure [260]. Exposure to cadmium species in the environment increases the risk of lung, kidney, prostate, pancreatic, breast, and urinary system cancer. The mechanisms by which cadmium promotes carcinogenesis are primarily dependent on oxidative stress coupled with the inhibition of antioxidants [40]. It also promotes lipid peroxidation and alters DNA maintenance mechanisms [261]. 

It has been observed that people who live in Japan in areas with soil contaminated by Cd developed an “Itai-Itai” disease, which is a severe kidney and bone syndrome with fractures and bone deformation [262], as Cd affects vitamin D and calcium assimilability [263]. Other diseases commonly associated with Cd exposure are anemia [264], diabetes [265], and osteoporosis [263]. Cd also has deleterious effects on the male reproductive system. It alters the migration of germ cells in the testis and reduces sperm count and motility [266]. Cd also affects the cardiovascular system, the lungs, the brain, the pancreas, and the adrenal glands [267]. Prevention and treatment methods depend on cadmium chelating agents, such as Ca-EDTA [268], meso-2,3-dimercaptosuccinic acid (DMSA), and its lipophilic alkyl monoester MiADMSA [269,270]. Moreover, antioxidant vitamins A, C, and E, as well as selenium, zinc, and magnesium supplementation, have been proven to detoxify cadmium [271]. 

In rats, Cd has been shown to be taken up from the intestinal tissues by metal transporters normally involved in Cu, Fe, and Zn uptake [272]. Divalent metal transporter 1 (DMT1) and ferroportin 1 (FPN1) and their human homolog Solute Carrier family 11 member 2 (SLC11A2) and SLC40A1, respectively, are involved in iron adsorption, while at low Fe level the Cd uptake increases [241,273]. Hereditary hemochromatosis (HH) is associated with SLC11A2 overexpression in the duodenum, which leads to iron accumulation [222]. The impairment of SLC40A1 function, leading to increased iron levels in specific brain regions, causes multiple neurodegenerative disorders, such as Parkinson’s disease, Huntington’s disease, and Alzheimer’s disease. Lower SLC40A1 expression is correlated with aggressive breast cancer [226]. The overexpression of SLC11A2 has been linked to esophageal [223] and colorectal cancer [223,224]. In ovarian cancer, SLC11A2 functions as a biomarker and a possible therapeutic target [225]. The overexpression of the zinc transporter ZIP8 and its human homolog SLC39A8 has been found to correlate with Cd-sensitivity. It also plays a crucial role in the development of inflammation, and its knockdown is harmful to mitochondria and increases cell death [274]. Other members of the SLC39 family, such as SLC39A4 and SLC39A14, have also been shown to be involved in Cd uptake [272]. The loss of function in SLC39A4 causes acrodermatitis enteropathica, a disease-causing systemic zinc deficiency. SLC39A14 has been demonstrated to be responsive to interleukin 6 (IL-6) in an acute-phase reaction as it localizes in the plasma membrane of hepatocytes and leads to one of the classic acute-phase responses, namely hypozincemia in the liver [227]. 

Copper-transporting P-type ATPase, ATP7A, also transports cadmium [272]. Mutations in the ATP7A gene cause Menkes disease (MD), occipital horn syndrome (OHS), and distal motor neuropathy (DMN) [228]. MD, also called kinky hair disease, is an X-linked disorder. It results in the elimination of copper from all the tissues except for the liver. In the brain, copper levels are abnormally high. The symptoms of MD are seizures, psychomotor retardation, hypoglycemia, and representative kinky hair with hyperelastic skin, bone fractures, and aneurysms. OHS, also known as X-linked cutis laxa or Ehlers-Danlos syndrome type 9, is a less acute variant of MD. Neurologic symptoms can be absent while the most typical abnormality is the calcification of the trapezius and sternocleidomastoid muscles at their attachments to the occipital bone. In DMN, the atrophy and weakness of distal muscles in the hands and feet are observed [228].

## 6. Summary

The ubiquitous environmental presence of metalloids and heavy metals, especially arsenic, antimony, and cadmium, poses a serious threat to human health. In the course of evolution, cells developed multiple mechanisms of protection against these toxins. Those mechanisms involve membrane proteins, which are crucial for the transport of stressors across biological membranes. Thus, in order to minimize the adverse activity of arsenic, antimony, and cadmium, cells rapidly, specifically, and selectively regulate the protein composition at multiple stages. These include the regulation of the transcription of crucial genes encoding membrane proteins [88,89,91,125,126], the post-translational regulation of the activity of these proteins [144,145,146,147], and the degradation of the misfolded ones or those involved in the uptake of heavy metals and metalloids [148,193,209].

Although knowledge of the topic is crucial to preventing the effects of exposure to arsenic, antimony, and cadmium, studying them in human cells presents a major challenge for scientists. A great advantage is provided by research on yeast, which is an excellent model for studying the processes occurring in the cells of higher eukaryotes [18]. The results obtained through these studies could possibly be employed to devise and improve therapies against membrane protein-related diseases associated with exposure to arsenic, antimony, and cadmium [214,217,226]. On the other hand, the compounds of these toxic elements have a chance to be used against serious human diseases, such as cancer or parasite-caused leishmaniasis. For instance, given the glycolysis-centered metabolism of cancer cells, they are known to overproduce glucose transporters of the GLUT family, which are homologous to yeast HXT transporters and constitute a major pathway of arsenic uptake [105,106,275]. Moreover, arsenicals are known inhibitors of these transporters and interact almost irreversibly with their active centers [276,277,278]. Nevertheless, the future of possible therapies depends on a thorough understanding of the processes responsible for the regulation of arsenic, antimony, and cadmium transporters; thus, further research in this area is necessary. 

## Figures and Tables

**Figure 1 ijms-25-04450-f001:**
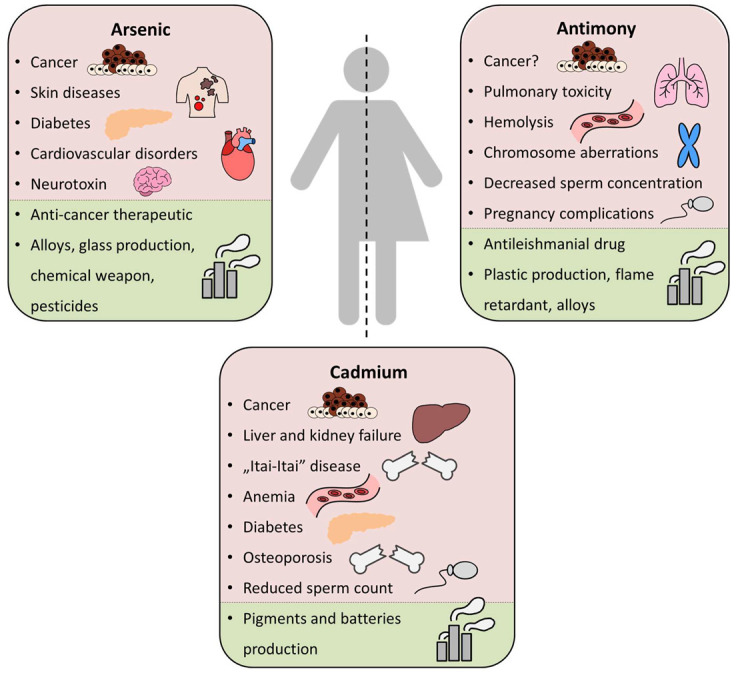
The effect of arsenic, antimony, and cadmium on human health and life. Light red indicates the negative effects and diseases caused by poisoning with particular metals/metalloids on human health. Light green indicates the positive use of metals/metalloids in industry and drug therapies.

**Figure 2 ijms-25-04450-f002:**
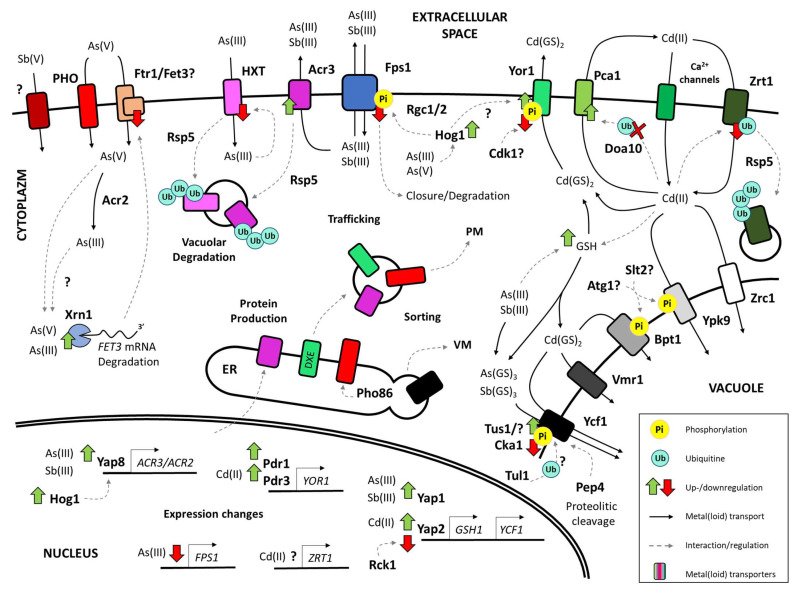
Schematic representation of metal(loid) detoxification pathways in yeast and the selected mechanisms of regulation of the membrane transporters in response to metal(loid) stress. The main pathways and mechanisms of multilevel regulation of membrane proteins at distinct cellular compartments. ER—endoplasmic reticulum; VM—vacuolar membrane; PM—plasma membrane; Pi—phosphate group; Ub—ubiquitin; GSH—reduced glutathione; DXE—ER exit signal. Membrane transporters depicted as colored rounded rectangles—vacuolar in grayscale, cadmium plasma membrane transporters in green shades, arsenic, and antimony plasma membrane transporters in reddish/pink shades, bidirectional Fps1 channel as blue. Presented regulation steps involve expression changes, protein production, sorting, and trafficking, as well as post-translational modifications, ubiquitylation, and vacuolar degradation. Black arrows indicate metal(loid) form changes and transport, and dashed gray arrows indicate interactions or regulatory aspects of factors impacting the activity of the selected transporters. Thick green and red arrows represent the up- and downregulation of protein level/function. More elaborate explanation of all depicted regulation pathways can be found in the text.

**Table 2 ijms-25-04450-t002:** Human metal(loid) transport-related proteins and diseases associated with their dysfunction.

Membrane Transporter	Aberration	Disease	References
**Arsenic/Antimony**			
ABCA1	global deficiency, loss of function	Tangier disease, coronary heart disease (CHD), type 2 diabetes (T2D), thrombosis, age-related macular degeneration (AMD), glaucoma, viral infections development, brain injury, Alzheimer’s disease, Parkinson’s disease	[213,214]
ABCB1	overexpression	ovarian cancer with multidrug resistance	[215]
GLUT1	downregulation	GLUT1-deficiency syndrome	[216]
AQP9	overexpression	chronic liver injury (CLI), systemic inflammatory response syndrome (SIRS)	[217,218]
AQP9	downregulation	hepatocellular carcinoma (HCC), polycystic ovary syndrome (PCOS), hyperandrogenism, altered sperm maturation and storage	[219,220,221]
**Cadmium**			
SLC11A2	overexpression	hereditary hemochromatosis, esophageal cancer, colorectal cancer, ovarian cancer	[222,223,224,225]
SLC40A1	disrupted function	Parkinson’s disease, Huntington’s disease, Alzheimer’s disease	[226]
SLC40A1	downregulation	aggressive breast cancer	[226]
SLC39A4	loss of function	acrodermatitis enteropathica	[227]
ATP7A	gene mutations	Menkes disease (MD); occipital horn syndrome (OHS), distal motor neuropathy (DMN)	[228]

ABCA1/ABCB1—ATP-binding cassette protein A1/B1; AQP9—Aquaporin 9; SLC11A2—Solute Carrier family 11 member 2; ATP7A—P-type ATPase A7.

## Data Availability

No new data were created or analyzed in this study. Data sharing is not applicable to this article.

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
