# Peer review of "Multilevel Regulation of Membrane Proteins in Response to Metal and Metalloid Stress: A Lesson from Yeast"

_ijms, 2024, doi:10.3390/ijms25084450_

Round 1

Reviewer 1 Report

Comments and Suggestions for Authors

The authors have collected a plethora of info but have made no effort to process it. In other words, at the moment the work is a well-organized list of published data.

The simplicity and essentiality of the two figures, representing commonly known facts,  are the proof.  The 'interplay' between the various factors (activators, transporters, ...) should be presented in a more accurate and elaborated way and enriched with schemes depicting the MULTILEVEL REGULATION promised in the title. 

Therefore, in my opinion,  the manuscript cannot be published until amended with more explanatory texts and schemes.

Author Response

Thank you for indicating of figures to be too generic. We heavily updated Figure 2 to more adequately depict the multilevel regulation of metal and metalloid transporters, summarizing major regulatory points production/activity. We believe that it better suits the need to explain those complex pathways after the revision. Please note that many regulatory pathways are still ambiguous, and this review intended to summarize the data, rather than complex analysis, which may be a topic of a future review. We hope that this updated figure is sufficient, given the short time for all the changes.

List of changes:

  • Lines 125-127 Figure 2A and 2B were replaced with new, Figure 2 which is more complex and presents more data on multilevel regulation of metal and metalloid transporters
  • Figure 2A and Figure 2B descriptions were changed to “Figure 2” due to the Figure 2 changes in lines 60, 64, 197, 245, 249, 269, 279, 297, 307, 332, 347, 362, 367, 373, 403, 508, 710
  • Lines 128-142 – A new updated description of Figure 2 was added
  • Line 506 “by Pep4 protease” was added to be more specific and consistent with the new figure 2
  • Lines 23-24 “analyzed” was changed to “present” to better suit the text and “orthologues” was added

Reviewer 2 Report

Comments and Suggestions for Authors

This manuscript provides a comprehensive review of yeast responses to As, Sb and Cd.  It also makes connections to responses in other species, primarily humans.  Several points could use further explanation.

Lines 87, 622-628.  While it is true that AsIII is generally more potent than AsV as stated (line 87), the facile intracellular conversion of AsV to AsIII is not mentioned.  This phenomenon confounds interpretations of differences in response to treatment with the two forms separately.  Short treatments are more likely to show differences than extended treatments where much conversion takes place.  The reader must wonder whether effects of both forms on the iron uptake system could reflect interconversion (lines 622-628).

Lines 98-99.  AsIII has high affinity for vicinal thiols and can form stable products, but not for solitary thiols, to which the binding is transitory.  Thus, binding to Zn finger elements in proteins has been seen as a major target (e.g., PARP-1; doi: 10.1016/j.taap.2017.05.031), whereas binding to solitary protein thiols is likely to be swamped out by the high concentration of glutathione in cells.

An explanation of the color scheme in Figure 2 would be helpful.

Lines 120-125.  The review makes no mention that AsIII and SbIII are essentially equivalent in numerous effects in human epidermal cells (doi: 10.1038/s41598-020-59577-0).

Lines 127-128, 131-132.  That a biological function for Cd in higher organisms has not been found is also true for As and Sb.  The latter also induce oxidative stress and HMOX1.

Line 430.  By “detoxify”, does the text mean simply “transport out of the cell”?  Detoxify suggests something beyond that.

Line 559.  The review gives no explanation for sensitivity differences between yeast and mammalian cells.  Thus the use of 1 mM AsIII for yeast cells will be startling for readers unaware of their apparent insensitivity but who are aware 1 mM would be quickly fatal for mammalian cells.  They will wonder how to rationalize comparing mechanisms in the two disparate types of organism.

Lines 712-713, 782.  The reader would appreciate more specific information than the vague “altered” in discussing effects.  Examples include (a) better to reveal whether the transcription went up or down (lines 711-712), and (b) whether “decreases” rather than “abnormalities” in SLC function are responsible (line 782).

Minor point – Note misspelling of chemical (chemiacal) under Arsenic in Figure 1.

Author Response

Thank you for your very constructive comments and a lot of suggestions for changes and clarifications. We think, we addressed the majority of them, which in our opinion positively impacts the clarity of this review for the reader. We added information on As and Sb toxicity in epidermal cells, thiol, and zinc-finger binding associated toxicity. We clarified the possible reduction of As(V) in the case of the Fet3/Ftr1 iron uptake system, highlighted some causes underlying the differences in yeast sensitivity to arsenic/antimony, and more.

The exact list of changes addressing the presented suggestions:

  1. Lines 87, 622-628. While it is true that AsIII is generally more potent than AsV as stated (line 87), the facile intracellular conversion of AsV to AsIII is not mentioned.  This phenomenon confounds interpretations of differences in response to treatment with the two forms separately.  Short treatments are more likely to show differences than extended treatments where much conversion takes place.  The reader must wonder whether effects of both forms on the iron uptake system could reflect interconversion (lines 622-628).
  • Lines 675-680 an additional comment was added on the uncertainty of arsenic form which is causing the effect of Fet3/Ftr1 downregulation
  1. Lines 98-99. AsIII has high affinity for vicinal thiols and can form stable products, but not for solitary thiols, to which the binding is transitory.  Thus, binding to Zn finger elements in proteins has been seen as a major target (e.g., PARP-1; doi: 10.1016/j.taap.2017.05.031), whereas binding to solitary protein thiols is likely to be swamped out by the high concentration of glutathione in cells.
  • Lines 113-118 additional information on thiol binding in zinc-finger domains was added
  1. An explanation of the color scheme in Figure 2 would be helpful.

  • Lines 128-142 – A newly updated figure as well as a description of Figure 2 was added, including an explanation of the color scheme
  1. Lines 120-125. The review makes no mention that AsIII and SbIII are essentially equivalent in numerous effects in human epidermal cells (doi: 10.1038/s41598-020-59577-0).
  • Lines 122-124 description of arsenic toxicity to epidermal cells was added
  • Lines 154-155 added information on antimony's influence on epidermal cells
  1. Lines 127-128, 131-132. That a biological function for Cd in higher organisms has not been found is also true for As and Sb. The latter also induce oxidative stress and HMOX1.

  • Line 159 “is a heavy metal that” added, line 160 “similarly to metalloids, arsenic, and antimony,” was added to address the comment that indeed no physiological function for arsenic and antimony has been discovered as well, which was previously mentioned in the text only for cadmium
  • Line 165 “like HSP70” and “(HMOX1) was added to indicate the influence of cadmium – cadmium-depended heat-shock activation as well as oxidative stress-inducing heme oxygenase HMOX1
  1. Line 430. By “detoxify”, does the text mean simply “transport out of the cell”?  Detoxify suggests something beyond that.
  • Lines 474-475 unable to “detoxify” was changed to “export the” and “out of the cell” was added to better portray the nature of Acr3 activity
  1. Line 559. The review gives no explanation for sensitivity differences between yeast and mammalian cells.  Thus the use of 1 mM AsIII for yeast cells will be startling for readers unaware of their apparent insensitivity but who are aware 1 mM would be quickly fatal for mammalian cells.  They will wonder how to rationalize comparing mechanisms in the two disparate types of organism.
  • Lines 79-86 clarification of the sensitivity differences was added
  1. Lines 712-713, 782. The reader would appreciate more specific information than the vague “altered” in discussing effects.  Examples include (a) better to reveal whether the transcription went up or down (lines 711-712), and (b) whether “decreases” rather than “abnormalities” in SLC function are responsible (line 782).
  • Table 2 ABCA1 Aberration was changed, now it specifies the deficiency or function loss instead of less specific altered expression
  • Line 766 “Globally altered ABCA1 expression” was changed to “Global deficiency of ABCA1” which is more specific on the cause of the Tangier disease
  • Lines 837-838 “Abnormalities in SLC40A1” was changed to “Impairment of SLC40A1 function, leading to increased iron levels in specific brain regions” for clarification of the nature of changes leading to the diseases mentioned later in the sentence

  1. Minor point – Note misspelling of chemical (chemiacal) under Arsenic in Figure 1.
  • Line 43-44 Misspelling was changed in “chemical” Figure 1

Reviewer 3 Report

Comments and Suggestions for Authors

Dear Editor and Authors

The manuscript presents a review on the multilevel regulation of membrane proteins associated with exposure to metals and metalloids based on research with yeast

The approach is interesting and the manuscript is well written. However, there is a lack of description of the search methodology for articles included in the review, including keywords, search bases and article inclusion and exclusion criteria. I do not consider it a requirement that the review be systematic, but even a critical review must minimally present its search criteria. This way, the interested reader will be aware of the depth of research that resulted in the review.

Therefore, I consider the manuscript should undergo a major review before publication in IJMS.

Author Response

Thank you for the comment, and for pointing out the lack of the search criteria. This review intended to highlight and summarize the data of expression and regulation of previously selected metal and metalloid transporters which are enlisted in Table 1. To address this issue we added a small clarification on that the transporters were selected and that those transporters were used as keywords during the research.

  • Line 22 “selected” was added
  • Lines 23-24 “analyzed” was changed to “present” to better suit the text and “orthologues” was added
  • Line 88 further was added

Lines 88-92 additional explanation of the topic covered was added to better indicate the intention and literature chosen for this review

Round 2

Reviewer 3 Report

Comments and Suggestions for Authors

No comments.